# Photocatalytic Removal of Thiamethoxam and Flonicamid Pesticides Present in Agro-Industrial Water Effluents

Michalis K. Arfanis [1], George V. Theodorakopoulos [1,2], Christos Anagnostopoulos [3], Irene Georgaki [3], Evangelos Karanasios [3], George Em. Romanos [1], Emilia Markellou [3] and Polycarpos Falaras [1,*]

[1] Institute of Nanoscience and Nanotechnology, National Center of Scientific Research "Demokritos", Agia Paraskevi, 15310 Athens, Greece

[2] Inorganic and Analytical Chemistry Laboratory, School of Chemical Engineering, National Technical University of Athens, Zografou Campus, 9 Iroon Polytechneiou Str., Zografou, 15772 Athens, Greece

[3] Benaki Phytopathological Institute, 8 St. Delta Str., 14561 Kifissia, Greece

[*] Correspondence: p.falaras@inn.demokritos.gr; Tel.: +30-2106503644

**Abstract:** Pesticide residues, when present in agricultural wastewater, constitute a potential risk for the environment and human health. Hence, focused actions for their abatement are of high priority for both the industrial sectors and national authorities. This work evaluates the effectiveness of the photocatalytic process to decompose two frequently detected pesticides in the water effluents of the fruit industry: thiamethoxam-a neonicotinoid compound and flonicamid-a pyridine derivative. Their photocatalytic degradation and mineralization were evaluated in a lab-scale photocatalytic batch reactor under UV-A illumination with the commercial photocatalyst Evonik P25 $TiO_2$ by employing different experimental conditions. The complete degradation of thiamethoxam was achieved after 90 min, when the medium was adjusted to natural or alkaline pH. Flonicamid was proven to be a more recalcitrant substance and the removal efficiency reached ~50% at the same conditions, although the degradation overpassed 75% in the acidic pH medium. Overall, the pesticides' degradation follows the photocatalytic reduction pathways, where positive charged holes and hydroxyl radicals dominate as reactive species, with complete mineralization taking place after 4 h, regardless of the pH medium. Moreover, it was deduced that the pesticides' degradation kinetics followed the Langmuir-Hinshelwood (L-H) model, and the apparent rate constant, the initial degradation rate, as well as the L-H model parameters, were determined for both pesticides.

**Keywords:** thiamethoxam; flonicamid; insecticides; photocatalytic degradation; titanium dioxide; scavengers; solution pH

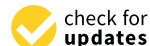



## 1. Introduction

At present, agricultural industry activities encompass the utilization of a plethora of pesticides, such as fungicides, herbicides and insecticides. Even if the pesticide usage improves the quality, and augments the productivity of, cereals, top fruits and vegetables, it also induces potentially harmful effects for the aquatic environment and human health [1]. Certain pesticides are recalcitrant organic molecules, non-biodegradable and resistant in removal by means of conventional wastewater treatment methods, eventually accumulating in soil and groundwater. Regarding the European legislations and watchlists [2,3], the impact and the hazards of the pesticides on environment and human health are constantly monitored and the database of the permitted pesticides is frequently updated.

As an example of a commonly detected substance in groundwater, thiamethoxam (TMX) an insecticide based on the natural toxin of neonicotinoid and was considered to be an environmentally benign compound when it was first introduced in crop protection [4]. This insecticide helps to deactivate the nervous system of several pests, such as whiteflies, aphids and micro-lepidoptera, thus protecting a wide range of crops, such as top fruits, vegetables, legumes, potatoes, barley, cotton and sunflowers [5,6]. However, TMX, as a

highly soluble and recalcitrant molecule, may gradually accumulate in aquatic environments and soil [7]. More recently, many countries have begun to face problems with bee populations due to TMX usage, implying its direct negative impact on the environment and food chain [4,8]. Toxicity examination revealed that TMX and its intermediates produced during disinfestation processes could harm algae and bacteria [9]. At present, TMX is non-approved in the European Union (EU) [2], while the Environmental Protection Agency (EPA) of the United States of America (USA) frequently reconsiders the ecological risk assessments and regulations for crops in consecutive publishable guidance, entitled "Thiamethoxam; Pesticide Tolerance".

On the other hand, many alternative and efficient pesticides with lower toxicity towards mammals, birds and fishes have been developed for agricultural use. For example, flonicamid (FND), a biodegradable pyridine carboxamide compound, is used in wheat, potatoes and citrus, acting against aphids, whiteflies and sucking insects through the inhibition of pests' feeding activity, without raising serious ecological and health concerns [10,11]. However, even if there is no unacceptable risk for humans, FND can be present in surface and groundwater, tea leaves and their extractions, or be poisonous for beneficial insects via contaminated honeydew [12–14]. The continuous discharge of FND in the environment could create potential risks for the agro-products consumption in the future; therefore, appropriate methods for detecting FND in human serum and urine in low quantities (~ng/L) have been already established [15,16].

Depending on the stability of pesticide residues, chemical substances remaining in the soil may reach the surface or groundwater. Although the EU has a strict regulatory framework for the authorization and waste management of pesticide residues, the development of novel removal techniques is essential as implementation tools. Conventional wastewater treatment methods, involving adsorption [17], chemical coagulation [18], membrane filtration [19], ion exchange resins [20] or other chemical systems [21], are costly and inefficient for their abatement [22]. Among the proposed solutions, advanced oxidation technologies (AOPs) based on heterogeneous photocatalysis with titania ($TiO_2$) are very promising treatment methods [23]. Titania is a non-toxic, low-cost and earth-abundant semiconductor, endowed with the capacity to treat water contaminated with pollutants by forming highly reactive and non-selective radical species under UV-A illumination [24,25]. Titania is a well-studied photocatalyst, which has been established as an appropriate candidate to be incorporated into pilot wastewater treatment plants of municipal, industrial or agro-industrial influents, aiming to eliminate a variety of organic pollutants, including pesticides, antibiotics, etc. [26–28].

In this work, the photocatalytic degradation of TMX and FND insecticides was investigated in order to evaluate the ability of titania, under a variety of experimental conditions, to decline these highly water-soluble pesticides and to avoid their leaching to the environment. To the best of our knowledge, this is the first time that such an extensive analysis on the photocatalytic performance of $TiO_2$ against TMX and FND has been implemented alongside an elaboration of the effects of pesticide dosage, solution pH and quenchers present during the photocatalytic process. The obtained results on these water contaminants are currently employed in the development and optimization of an innovative hybrid photocatalytic nanofiltration reactor prototype unit (PNFR), with the capability to recycle 15 m$^3$/day of real agro-wastewater [29].

## 2. Results

### 2.1. Photocatalytic Degradation and Reaction Kinetics of Pesticides

To begin, the kinetics of pesticides adsorption and photolysis were determined. In the absence of UV-A light (dark conditions), both the TMX and FND presented negligible adsorption onto the catalyst, even after 90 min. Considering these preliminary results, it was concluded that even 30 min of stirring under dark conditions was sufficient to reach an adsorption/desorption equilibrium. Subsequently, the molecules' photostability was also corroborated under UV-A illumination for a period of 90 min and it was found that

the pesticides' concentrations did not alter significantly in the absence of photocatalyst P25 (UV photolysis). In summary, the adsorption and photolysis effects should be considered as negligible during the photocatalytic processes; thus, no interferences with the interpretations of the photocatalysis results were expected.

The effect of the initial concentration on the photocatalytic performance of the commercial P25 for both pesticides was studied and the order of the photocatalytic degradation reaction was determined from experimental tests at concentrations of 1, 5, 10 and 20 ppm. Hence, the photocatalytic degradation kinetics of the polar and water soluble TMX and FND were examined, and the relevant results are presented in Figure 1. It has often been noticed that the rate of the heterogeneous photocatalytic degradation of dyes and pesticides follows the Langmuir-Hinshelwood (L-H) model [30], as described in detail in the supplementary information (SI). In brief, the calculated parameters derived from the L-H model are: the apparent rate constant ($k_{app}$, min$^{-1}$), the initial degradation rate ($r_{r,0}$, mg·L$^{-1}$·min$^{-1}$), the reaction rate constant ($k_r$, mg·L$^{-1}$·min$^{-1}$) and the adsorption constant of the reactant ($K_{LH}$, L·mg$^{-1}$).

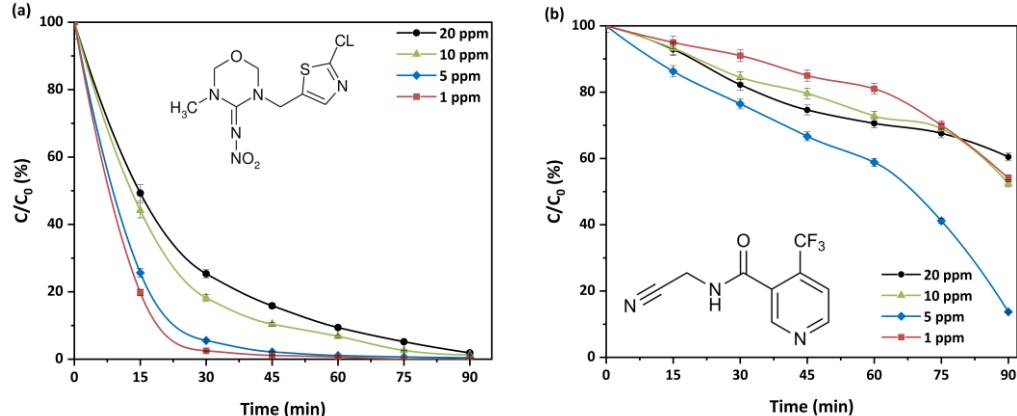

**Figure 1.** Degradation kinetics of photocatalytic degradation of (**a**) thiamethoxam (TMX) and (**b**) flonicamid (FND) using titania P25 at the studied concentrations (UV-A irradiation, 0.1 g/L TiO$_2$, natural pH, 25 °C).

A plot of the $\ln(C_0/C)$ versus time for the examined initial concentrations is displayed in Figure S1a,b and the linear regression slope is equal to the $k_{app}$ rate constant. The $k_{app}$ and $r_{r,0}$ values corresponding to these initial concentrations are summarized in Table 1 for both pesticides. As observed in Figure S1c, for both pesticides, the initial degradation rate increases with the increasing initial concentration, even at highly concentrated solutions, proving the excellent photocatalytic efficiency of the photocatalyst without having any sign of a saturated and finite performance.

**Table 1.** Apparent pseudo first-order rate constants and initial reaction rates calculated for the different initial concentrations of TMX and FND. The presented half-life reaction times for both pesticides have been calculated at the examined initial concentrations.

| | Thiamethoxam | | | | Flonicamid | | | |
|---|---|---|---|---|---|---|---|---|
| $C_0$ (mg·L$^{-1}$) | $K_{app}$ (min$^{-1}$) | $r_{r,0}$ (mg·L$^{-1}$·min$^{-1}$) | $t_{1/2}$ (min) | $t_{1/2}'$ (min) | $K_{app}$ (min$^{-1}$) | $r_{r,0}$ (mg L$^{-1}$ min$^{-1}$) | $t_{1/2}$ (min) | $t_{1/2}'$ (min) |
| 1.0 | 0.0998 | 0.091 | 6.81 | 6.95 | 0.0035 | 0.004 | 217.46 | 198.61 |
| 5.0 | 0.0848 | 0.397 | 8.56 | 8.17 | 0.0089 | 0.046 | 252.21 | 77.53 |
| 10.0 | 0.0479 | 0.368 | 11.10 | 14.48 | 0.0052 | 0.054 | 295.64 | 132.03 |
| 20.0 | 0.0399 | 0.849 | 16.47 | 17.38 | 0.0061 | 0.120 | 379.89 | 114.00 |

The $k_r$ and $K_{LH}$ values for TMX, calculated from the slope and intercept of the linear regression, were 1.0504 mg·L$^{-1}$·min$^{-1}$ and 0.1042 L·mg$^{-1}$, respectively, with a coefficient of

determination of $R^2 = 0.992$. As the initial adsorption rates of TMX in the dark could not be calculated from the experimental data due to the negligible adsorption, it can be concluded that the initial TMX photodegradation rates are much faster than the adsorption rates under dark conditions. In this context, the experimental results with TMX could only be reconciled with the Langmuir-Hinshelwood model by assuming light-induced changes of the photocatalyst surface, which may have a significant effect on the adsorption of the probe molecule. Similarly, the $k_r$ and $K_{LH}$ of FND were calculated to be 0.0576 mg·L$^{-1}$·min$^{-1}$ and 0.0577 L·mg$^{-1}$, respectively, with a coefficient of determination of $R^2 = 0.99$, using the parameters in Table 1, which are in accordance with first-order kinetics. In the case of the TMX, the rate constant was presented to be 18 times higher than that of the FND, verifying the more recalcitrant nature of the FND pesticide. Moreover, the titania photocatalyst showed a higher tendency of adsorbing the TMX more strongly compared to the FND, as the $K_{LH}$ was almost double (0.1042 and 0.0577 L·mg$^{-1}$, respectively). Finally, the similar trend for the $K_{LH}$ of the FND with the respective to the TMX concludes that the assumption of light-induced changes on the surface properties of the photocatalyst is valid and that this asset has a significant effect on the adsorption of the pesticide molecule during irradiation.

As observed in Figure 1a,b, the data points of all the experimental runs for both pesticides were well fitted, employing an exponential decay model for the TMX, although for the FND, a more linear decay was observed. In addition, the neonicotinoid TMX presented high photodegradation rates, reaching complete removal after 60 min for the diluted solutions (1 and 5 ppm) and 90 min for the more concentrated ones, with no toxic by-product residuals in the solution. As expected, the FND was more resistant to photocatalysis, reaching removal efficiencies ~48% after the photocatalytic experiments, with the exception of the concentration of 5 ppm [31].

Furthermore, the half-life time of the reaction was calculated for the reaction rate of a pseudo-first order kinetics approximation, as it a valuable parameter in order to estimate the reaction rate [32]. In particular, at the half-life time $t_{1/2}$ of the reaction, where the concentration is half of the initial (C = 0.5·$C_0$), this time is calculated by the following equation:

$$t_{1/2} = 0.5 \cdot C_0/k_r + \ln2/k_r \cdot K_{LH} \tag{1}$$

In addition, for reactions exhibiting pseudo first-order kinetics, the half-life time based on the $k_{app}$ can be derived from the following equation:

$$t_{1/2}' = \ln2/k_{app} \tag{2}$$

When Equation (1) is true, the estimated values of the half-life time for different initial concentrations would be the same as those obtained from the observation (Equation (2)) [33]. The values of $t_{1/2}$ and $t_{1/2}'$ are summarized in Table 1, including the results obtained for both of the studied pesticides. As observed in this table, the half time of the TMX is much shorter compared to the corresponding values of the FND, implying that the TMX degradation has a faster rate. In addition, the evaluation of these values presented a difference between $t_{1/2}$ and $t_{1/2}'$, which became significant with a rise in the initial pollutant concentration. This trend, which was smoother in the case of the TMX solution, could be elucidated by the intermediates' formation [32,33], which could be adsorbed competitively on the photocatalyst, leading to the retardation of the kinetics. Hence, this effect becomes more prominent as the initial concentration of the pesticides increases, causing the generation of more intermediates, which in turn leads to the decline in the pesticides' degradation. Overall, the L-H model satisfactorily approves the TMX degradation, while the model cannot respond accurately to the degradation of the recalcitrant FND pesticide, especially in higher concentrations.

Figure 2 shows the photocatalytic degradation percentage and total organic carbon concentration (TOC) removal for the TMX (10 ppm) and FND (10 ppm) pollutants, following the UV-A irradiation of aqueous solutions for 90 min, in the presence of titania P25. Thus, the degree of their mineralization was determined, and the effectiveness of

the photocatalytic process was comparatively evaluated. It can be seen that after 90 min under UV-A irradiation, the carbon content percentage drops to 76% and 53.5% for the TMX and FND, respectively. This means that 24% of the TMX was mineralized after 90 min irradiation time, although the degradation for TMX was almost complete, probably due to the presence of intermediate organic products. Thus, the observed delay in the mineralization process with respect to the corresponding photocatalytic degradation degree is expected, as the recalcitrant parent compounds should be first converted to intermediate fragments, which undergo a progressive transformation leading to $CO_2$ and inorganic species as final products, until the complete removal of the organic load after ~4.5 h (Figure S2). Moreover, the degradation of the FND parent closely follows its direct mineralization in contrast to the TMX parent, while for both pesticides, mineralization presents a linear dependence with time (Figure S2). However, despite the presence of organic matter in the photocatalytically treated pesticide solutions indicated by the TOC analysis, no toxic by-products were detected, with the exception of traces (<0.0004 ppm) of clothianidin and 4-(Trifluoromethyl)nicotinoyl glycine degradation products of the TMX and FND, respectively [10,34]. Even if the analytical examination by HPLC-MS was limited only to expected toxic by-products of FND and TMX, these results clearly demonstrate the ability of the $TiO_2$ photocatalyst to degrade hazardous contaminants and its potential use in wastewater purification technologies.

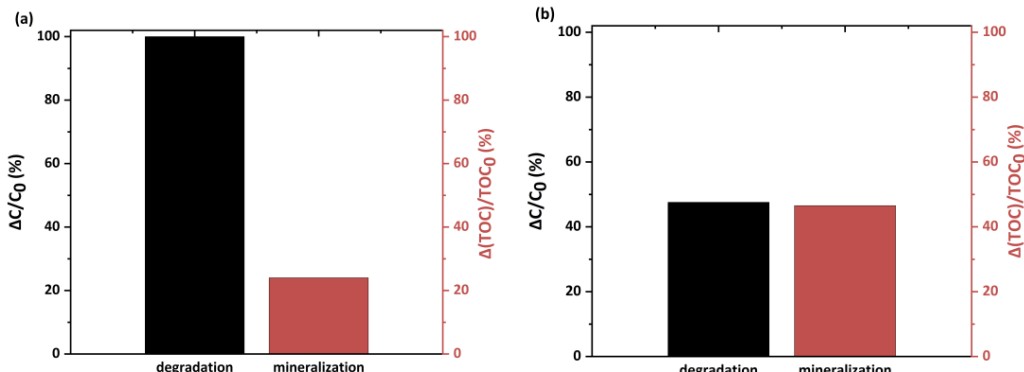

**Figure 2.** Photocatalytic degradation and TOC removal of TMX (**a**) and FND (**b**) using titania P25 (UV-A irradiation for 90 min, 10 ppm pesticides concentration, 0.1 g/L $TiO_2$, natural pH, 25 °C).

### 2.2. Effect of Solution pH

As the pH of real water varies relatively to the climate conditions and the type of effluents, the affinity between the organic molecules and the surface of the photocatalyst is highly correlated with the pH of the water matrix. In this work, we have considered that it is extremely important to conduct a thorough investigation of the effect of the pH on the photodegradation kinetics of TMX and FND. Regarding the catalyst, $TiO_2$ is a chemically stable semiconductor exhibiting a point of zero charge close to 6.2, meaning that it can be positively charged below pH = 6.2, or negatively above this value [35]. On the contrary, the pesticides could only be charged negatively if polarization occurs, while there are some studies reporting that the half-life of TMX is reduced in a strong alkaline environment [7,36]. The experiments were performed at a concentration of 10 ppm for both pesticides, whereas the pH was selectively adjusted by the addition of HCl (0.1 M) and NaOH (0.1 M). It should be noted that the choice of 10 ppm allows for the tracking and examination of the degradation alterations with time, regardless of the kinetics enhancement or hindering with pH adjustment. Moreover, the pH values of the starting TMX and FND solutions were 5 and 7, respectively. Therefore, the respective experiments were performed with no addition of HCl or NaOH.

Starting with TMX, the adsorption experiments in the dark showed that there are not any substantial alterations of the $TiO_2$ adsorption capacity with the pH (Figure 3a). Despite the changes in the $TiO_2$ surface charge, this behavior was expected, as TMX has

no dissociation constant ($pK_a$) [6]. Indeed, the TMX concentration was the same after the adsorption-desorption equilibrium, independently of the pH value (not shown), denoting that the changes in the photocatalytic degradation rate (with pH) should only be attributed to the variation in the active species concentration in the solution. As shown in Figure 3a, the photodegradation of TMX was similar for the experiments performed at pH = 7 and pH = 5 (the initial pH value of the 10 ppm solution), reaching almost 100% removal after 90 min under UV-A illumination. The reaction kinetics were decreased at pH = 9, although the TMX concentration was eventually zeroed. The increase in the $OH^-$ concentration in the alkaline solution shifts the reaction equilibria to the products, thus favoring the generation of hydroxyl radicals onto the photocatalyst and promoting a higher photo-quantum yield and protonation level of the excited pesticide molecules, as proposed by Liang et al. [37]. On the other hand, the acidic environment (pH = 3) hampered the removal efficiency significantly, which did not surpass the value of 40%. Yang et al. proposed, as a possible explanation, the protonation of TMX (the contained sulfur and nitrogen atoms can be protonated under these conditions); therefore, coulombic repulsion forces occur among the molecule and the positively charged titania (catalyst's surface), and the photocatalytic activity is reduced [7].

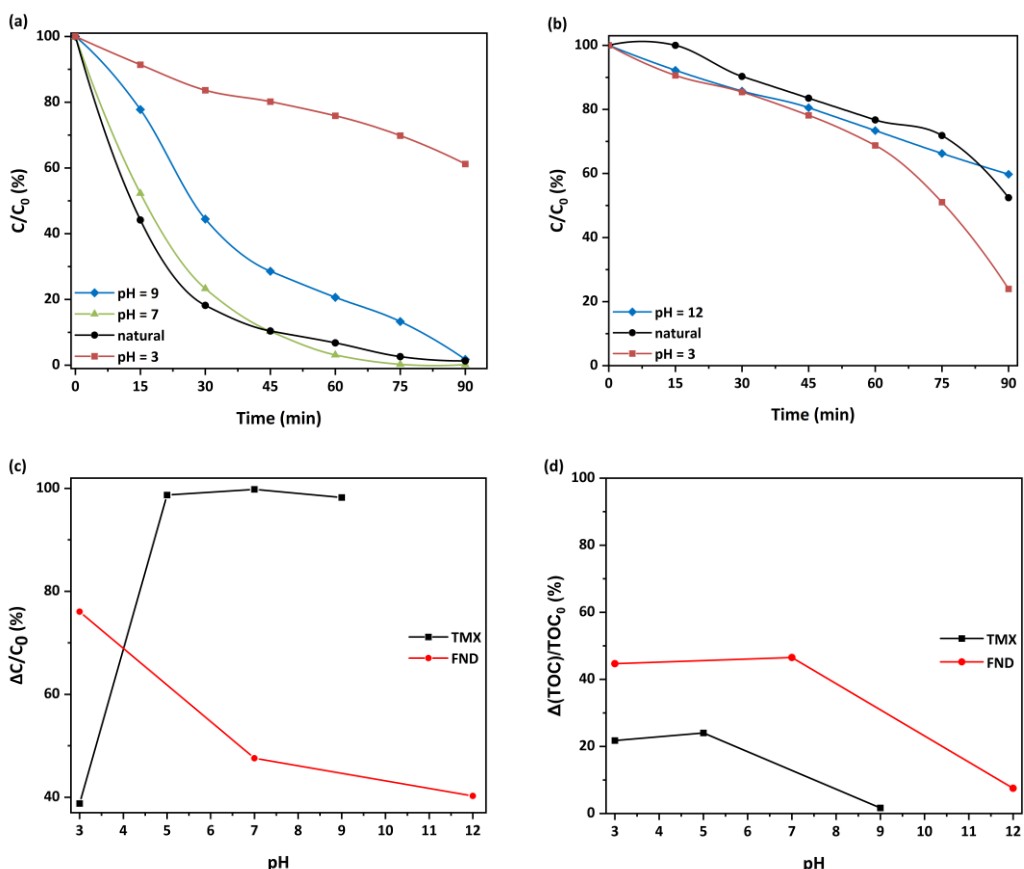

**Figure 3.** Effect of solution pH on the photocatalytic removal of (**a**) thiamethoxam (TMX) and (**b**) flonicamid (FND); (**c**) Comparative representation of medium pH effect on photocatalytic efficiency; and (**d**) total mineralization for both pesticides after 90 min UV-A irradiation.

Concerning the removal of FND from the solutions of varied pH (Figure 3b), the pesticide $pK_a$ value is close to 11 [38], so its polarization above that limit could affect the photocatalytic reaction. Nevertheless, when the pH was fixed to 12, the photocatalytic efficiency was not altered significantly compared to the performance of the initial solution (pH = 7). In general, both the FND and titania surface are negatively charged in such a medium, so the electrostatic repulsion forces generated among them are expected to restrict

the photocatalytic process. It is possible that the excess of $OH^-$ anions in the solution generated a high number of $^{\bullet}OH$ reactive radicals, which compensated the negative effect of electrostatic repulsion and, thus, the photocatalytic effect remained intact. Surprisingly, the acidification of the solution improved the FND degradation, reaching ~80% removal, although the FND molecule was not polarized and the sorption abilities of the positively charged $TiO_2$ remained insignificant (not shown). In this context, a possible reason for the achieved enhancement of the photocatalytic activity could be related to the easiest and faster dissociation of water molecules into hydroxyl radicals and the higher oxidation potential of the generated hydroxyl radicals at a lower pH in comparison to that at a higher pH [39].

As a result, the pH control can be considered to be a very important parameter for the pesticides' removal and its effect was clarified in each case (Figure 3c). The natural pH (pH = 5) fits the neonicotinoid pesticide better, whereas the solution acidification decreases the degradation rate. Instead, the FND degradation rate was noticeably improved in the acidic environment (Figure 3c), surpassing the photocatalytic efficiencies in the natural and alkaline media. Nevertheless, it should be also considered that the pH does not only affect the degradation of the target molecules during photocatalysis, but also the photodecomposition of their by-products (Figure 3d). The determination of the TMX and FND mineralization efficiencies in the acidic medium demonstrated that the mineralization was not altered compared to the initial pH at the same time intervals. Regardless of the observed photodegradation rates of the parent pesticides and the competitive phenomena among the initial compound and their by-products, the photogenerated reactive species can still attack and reduce the organic load. In contrast, the solution's pH adjustment to alkaline totally hindered the mineralization of the pesticides; even if the P25 photocatalyst broke up the target molecules to intermediate fragments, these fragments were not degraded further. It is possible that repulsion occurs between the organic substances and titania surface under the applied conditions; therefore, the species' activity might be impeded.

### 2.3. Effect of Additives as Quenchers

In addition to the pH variations, the photocatalytic processes could be either intensified or deteriorated by the presence of other additives in the contaminants' solutions. The addition of inorganic anions increases the removal efficiency of the catalysts if the anions act as oxidation agents, leading to an improved separation of the photogenerated charged carriers and prolonged holes lifetime, or if they are able to generate extra reactive oxidative species in the solution [7]. On the other hand, if the inorganic anions are accumulated on the photocatalysts' surface or if they quench the reactive species, then the $TiO_2$ is deactivated, and the efficiency attenuates [7]. When metal ions are added, photocatalysis could be more effective as metals act as an electron sink and/or additional catalyst, although hindering effects cannot be excluded [6]. The insertion of organic additives is more complex for the ROS photoactivity because competitive phenomena occur among the target pollutants and the additives [40].

To better understand the photocatalytic degradation mechanism, radical trap experiments were performed, employing the appropriate additives as quenchers [41], with high affinity in relation to the photocatalytic reactive species, such as isopropanol (IPA), potassium iodide (KI), benzoquinone (BZQ) and potassium bromate ($KBrO_3$). The obtained results (Figure 4a,c) reveal that both photocatalytic oxidation and reduction pathways can be equally considered as the reaction mechanism of TMX degradation. In particular, the addition of IPA, which is the scavenger consuming hydroxyl radicals, has the most negative effect during the photocatalytic degradation of TMX. On the other hand, KI (potassium iodide) mostly interacts with photogenerated holes, which are the charge carriers responsible for the photo-oxidation of the adsorbed water and the concomitant production of the hydroxyl radicals. It is thus reasonable that KI also has a significant negative impact on the pesticide photodegradation efficiency. Similarly, the photocatalytic degradation of TMX was adversely affected when BZQ was used as a superoxide radical quencher, implying that the superoxide radicals were also participating in the process. In contrast, the use of

KBrO$_3$ as an electron scavenger led to slightly improved photocatalytic degradation. This tendency has also been observed in the literature by Mir and co-workers [6], who proposed that KBrO$_3$ enhances the e$^-$/h$^+$ charge carriers' separation by acting as an electron acceptor. In fact, the excess of positive charged holes could now either degrade the TMX directly or generate the necessary hydroxyl radicals, leading to the enhanced photocatalytic oxidation efficiency of TMX under these conditions.

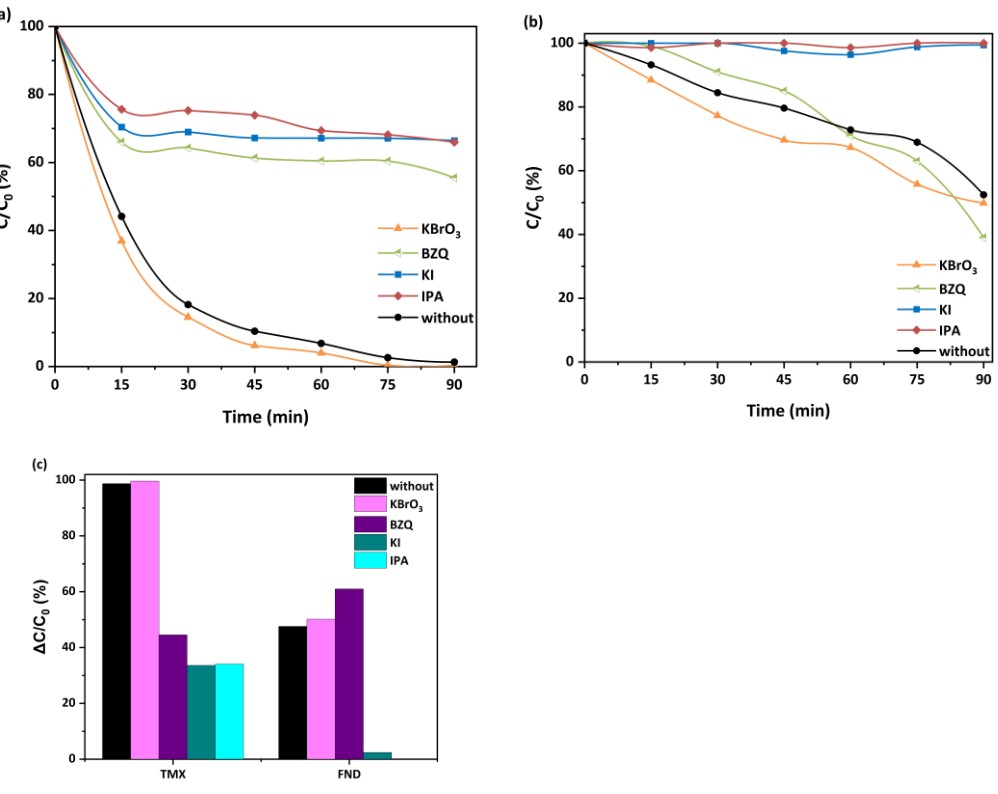

**Figure 4.** Radical trap experiments of (**a**) thiamethoxam (TMX); and (**b**) flonicamid (FND) in the presence of appropriate scavengers: IPA for $^\bullet$OH quenching, KI for h$^+$, BZQ for O$^{2\bullet-}$ and KBrO$_3$ for e$^-$; (**c**) Scavengers' effect on the photocatalytic degradation performance for both pesticides.

Thereafter, the main degradation mechanism for the FND was also elucidated through trap experiments (Figure 4b,c). The results showed a clear dependence of the pesticide removal under the photocatalytic oxidation pathways, while the trapping of the reductive species increased the photocatalytic efficiency. Specifically, the addition of IPA and KI entirely hindered the photocatalytic processes against the pesticide, clearly demonstrating that the FND degradation could not occur in the absence of hydroxyl radicals and holes, respectively. On the other hand, the removal rate of the FND in the presence KBrO$_3$ (electron acceptor) was enhanced with time. This observation indicates that the photogenerated electrons, which negatively affected the photocatalytic degradation pathways, were now quenched by the respective scavenger and could not participate in the recombination processes. Thus, the observed FND behavior arises from rather limited electron-hole recombination and the main reactive species are holes and/or hydroxyl radicals, which are both able to oxidize the FND pollutant.

## 3. Discussion

Photocatalytic experiments were also performed in the presence of both pesticides to access the potential of the titania photocatalyst in more realistic conditions. The results of the photocatalytic degradation and carbon removal (TOC) for the mixture (5 ppm of TMX and 5 ppm of FND) after 90 min are shown in Figure 5, and their comparison with

those of Figure 1a,b indicates that the photocatalytic kinetics and removal percentage in the mixture is almost identical to the performance achieved in the single pollutant experiments. Indeed, the excellent photocatalytic efficiency of the titania catalyst toward TMX was preserved. Regarding the FND, the photocatalytic degradation reaction remains relatively slow and proceeds with a significantly low rate, until the acceleration after 60 min, as observed in the aforementioned isolated experiments. Furthermore, the TOC analysis enabled the calculation of the mineralization degree of the pesticide mixture. As such, the mineralization efficiency in the mixture is compared to the average of the mineralization efficiencies achieved in the single solute experiments. In this context, a first observation reveals that the mixture mineralization proceeds at a slower rate compared to the respective degradation rates. In particular, the carbon removal efficiency of the mixture reached up to approximately 31%, which adequately coincides with the theoretical efficiency of 35%, as produced by averaging the efficiencies achieved in the single solute experiments (dashed line in Figure 5).

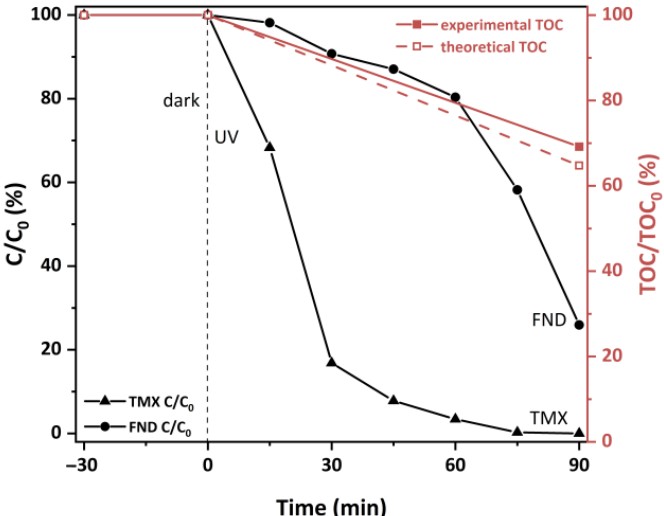

**Figure 5.** Kinetics of photocatalytic degradation and TOC removal of TMX and FND mixture (comparison with theoretical TOC removal of each pesticide individually) using titania P25 (UV-A irradiation, 5 ppm TMX and 5 ppm FND concentrations, 0.1 g/L $TiO_2$, natural pH, 25 °C).

The overall results confirm that the photocatalytic degradation of TMX and FND pesticides with P25 titania is feasible and can be effectively applied, regardless of the experimental conditions. The degradation and mineralization results of the TMX are in accordance with the literature data, where larger amounts of $TiO_2$ photocatalyst were used and where the photocatalysis was also combined with other advanced oxidation techniques [4,5,42]. The performance of Degussa P25 $TiO_2$ was compared with various photocatalysts in the form of slurry or immobilized on substrate from the literature reports and is presented in Table 2. Notably, for almost similar reaction conditions, the photocatalytic efficiency of Degussa P25 (with low light intensity) is approximately of the same level than that reported in the literature. Apparently, the photocatalytic conditions (catalyst loading, pH, light intensity) have a significant impact on the efficiency of a photocatalyst. Among them, light intensity constitutes a decisive factor as it plays a significant role in photocatalytic degradation, determining the number of generated electron-hole pairs. However, the absence of a common roadmap during the investigation of the photocatalytic process makes the comparison between the literature works rather difficult.

**Table 2.** Photocatalytic performance of Degussa P25 TiO$_2$ photocatalyst against TMX and FND compared to other photocatalysts reported in literature.

| Photocatalyst | Catalyst Amount (g/L) | Pesticide/Concentration (ppm) | Light Intensity (mW/cm$^2$) | Removal Efficiency (%) | Reference |
|---|---|---|---|---|---|
| CuO | 1 | FND/75 ppm (pH = 2) | High-pressure mercury UVA lamp (125 W), n/a | 52.73% (COD), 2 h | [11] |
| ZnO | 0.75 | FND/75 ppm (pH = 2) | High-pressure mercury UVA lamp (125 W), n/a | 60.58% (COD), 2 h | [11] |
| ZnO | 2 | TMX/~110 ppm (natural pH) | 1.75 | 77%, 2 h | [22] |
| TiO$_2$ onto glass slides | 0.24 (~10 mg on each slide) | TMX/100 ppm (pH n/a) | 42 | 90.1%, 2 h | [4] |
| ZnO | 0.2 | TMX/0.1 ppm (pH = 7.1) | 8.5 | 97%, 2 h | [43] |
| g-C$_3$N$_4$-TiO$_2$@LMPET | 4.3 (130 mg PET fiber mat) | TMX/5.8 ppm (pH = 1) | Q-sun Xe-1 test chamber (solar irradiation), n/a | >97%, 3 h | [44] |
| TiO$_2$ | 0.1 | TMX/10 ppm (natural pH) | 0.5 | ~99%, 1.5 h | This work |
| TiO$_2$ | 0.1 | FND/10 ppm (natural pH) | 0.5 | ~48%, 1.5 h | This work |

In the case of FND, this is the first instance of a full examination, whereas the available studies to date mainly confirm the molecule's recalcitrant nature with COD techniques [11,27,45]. In summary, the reported photocatalytic results of the pesticides removal (both isolated and mixture) are very promising, rendering TiO$_2$ materials as excellent candidates for scale-up applications, targeting their integration in existing wastewater treatment units. In this context, dynamic experiments under continuous flow in lab-scale are in progress in order to prove their applicability in an upscaled hybrid photocatalytic nanofiltration membrane reactor (PNFR) and their feasibility for agricultural wastewater purification processes in the fruit industrial sector [46,47].

## 4. Materials and Methods

The commercial titania Evonik Aeroxide P25 was selected as the model photocatalyst for the photocatalytic removal of pesticides from the water matrix. The experiments were carried out in open vessel glass vials in batch-mode, using 10 mL of aqueous solutions of thiamethoxam (TMX, analytical standard, Sigma-Aldrich, St. Louis, MO, USA) or flonicamid (FND, analytical standard, Sigma-Aldrich, St. Louis, MO, USA). Depending on the experiment, the pesticide concentration varied between 1 and 20 ppm, while 1 mg of P25 (0.1 g/L) was suspended in the aforementioned solutions. In brief, the photodegradation process of the pesticides with P25 was accomplished under UV-A irradiation, inside a lab-made photoreactor. The pesticide concentration and total organic carbon (TOC, BioTector B3500, Hach, Loveland, CO, USA) were calculated using analytical methods (LC-MS/MS, Varian model 1200 L, Agilent Technologies, Foster City, CA, USA) and a TOC analyzer, respectively, and then the respective photocatalytic and mineralization efficiencies were estimated. The effect of the solution's pH was evaluated, and trap experiments through the addition of the appropriate additives were also conducted to expose the photocatalytic degradation mechanism. The photocatalytic experimental procedures concerning the effect of the additives [25,41] and solution's pH, as well as the analytical procedure and evaluation [48,49], are described in detail in the supplementary information (SI).

## 5. Conclusions

In this study, the photodegradation of the thiamethoxam and flonicamid pesticides accumulated and frequently detected in the wastewater of the fruit industry, was achieved. In particular, TMX was totally removed from the solution after 90 min of UV-A illumination and an approximately 86% removal was achieved for the more recalcitrant FND pesticide (depending on the solution concentration) in the presence of the commercial titania photocatalyst Aeroxide P25. The observed degradation was accompanied by sufficient pollutant mineralization, while no toxic by-products were detected during the photocatalysis. In this extensive examination, the effect of the solution acidity and pollutant concentration on the photocatalytic efficiency were evaluated. Furthermore, the photocatalytic oxidation pathways were recognized as the main degradation mechanism for both the TMX and FND, where the photogenerated hydroxyl radicals were the most reactive species. These

results verify the highly photocatalytic performance of $TiO_2$ materials against pesticide pollutants with different physicochemical properties under various experimental conditions and promote their potential use in scale-up applications.

**Supplementary Materials:** The following supporting information can be downloaded at: https://www.mdpi.com/article/10.3390/catal13030516/s1, Table S1: Chromatography parameters with time for the pesticides' detection. Table S2: Chromatography parameters with time for the detection of FND metabolites. Table S3: Overview of the LCMS-MS parameters for the analytes investigated. Figure S1: Degradation kinetics of photocatalytic degradation of (a) Linear transform $\ln(C_0/C) = f(t)$ of thiamethoxam (TMX) and (b) flonicamid (FND) using titania P25 at the studied concentrations (UV-A irradiation, 0.1 g/L $TiO_2$, natural pH, 25 °C); Effect of the concentration on the initial degradation rate of both pesticides (c). Figure S2: Total mineralization efficiency of thiamethoxam (TMX) and flonicamid (FND) during the photocatalytic process (UV-A irradiation, 0.1 g/L P25 $TiO_2$, natural pH, 25 °C). Refs. [25,41,48,49] are cited in the Supplementary Material file.

**Author Contributions:** Conceptualization, G.E.R. and P.F.; Investigation, writing—original draft preparation, G.V.T. and M.K.A.; Validation, methodology, G.V.T., M.K.A., C.A., E.K. and I.G.; Project administration, E.M.; Resources, E.M. and P.F.; Supervision, writing—review and editing. G.E.R. and P.F. All authors have read and agreed to the published version of the manuscript.

**Funding:** This work was funded by the EC, Environment Program (EU: H2020 LIFE17 ENV/GR/000387 PureAgroH2O Project). The Green Fund is co-financing actions of the partner NCSR "Demokritos" in the frame of the implementation of the LIFE PureAgroH2O project.

**Data Availability Statement:** Not applicable.

**Acknowledgments:** P.F. acknowledges funding of this work by Prince Sultan Bin Abdulaziz International Prize for Water (PSIPW)-Alternative Water Resources Prize 2014.

**Conflicts of Interest:** The authors declare no conflict of interest. The funders had no role in the design of the study; in the collection, analyses, or interpretation of data; in the writing of the manuscript; or in the decision to publish the results.

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
