# Peer review of "Photocatalytic Removal of Thiamethoxam and Flonicamid Pesticides Present in Agro-Industrial Water Effluents"

_catalysts, doi:10.3390/catal13030516_

Round 1
Reviewer 1 Report
This work evaluated the effectiveness of the photocatalytic process to decompose two frequently detected pesticides TMX and FND in the water effluents of fruit-industry. The degradation kinetics were studied the experimental conditions, and satisfactory mineralization yields were accomplished. The experimental design of this study is reasonable. But I think it should be revised before publishing.
1. The format of thiamethoxam should be unified after the line 44, such as line 54 should be abbreviated as TMX.
2. In Figure 1, 3-5, Is the ordinate degradation efficiency or C/C0? Please make sure.
3. Table 2 should be merged into Table 1, which should be better.
4. If possible, the degradation of the target compound should be checked with actual water.
5. If possible, ESR spectra should be tested.
6. The influence of inorganic ion on the degradation efficiency should be discussed.
7. The format of references should be unified, please check the page number of the reference carefully.
Reviewer 2 Report
Major:
1. Abstract should emphasize the experimental findings, include results, numerical data
2. Introduction should give some statistics related to thiamethoxam and flonicamid, such as present levels found in soil and groundwater, what is the acceptable limit, and if found in living organisms what is the safety range and what will be the potential side effects.
3. Try to find more literature relevant to other potential treatment methods used for pesticide removal rather than generalizing the traditional methods.
4. In materials and methods, the method used for the LC-MS/MS was based on the previously published method or was it developed by the authors’ specific to their work. Kindly state the same accordingly.
5. For FND, what is the basis for limiting the experiment time to only 60 minutes? Please justify.
6. Figure 2: there is linearity in the adsorption of TMX at concentrations 1, 5, 10 and 20 ppm, whereas FND shows good adsorption at 5 ppm only. Then why choose 10 ppm concentration for both the pesticides, as Figure 1 clearly shows adsorption of TMX even at 20 ppm, whereas FND only at 5 ppm.
7. Please provide the TOC after achieving total mineralization (total time to achieve complete mineralization) as 90 min for TMX only achieved 24% mineralization. Similar for FND.
8. To support “effect of pH”, please provide the zeta potential data of TiO2 at test pH conditions.
9. Effect of solution pH: Why TOC was not performed during these experiments. It will provide good insights in knowing the effect of pH over mineralization as well.
10. Degradation of TMX and FND in mixture should also be performed at different pH conditions, as authors clearly stated the necessity of pH in the above section. Then only readers can clearly understand the importance of TiO2 in real-time degradation of TMX and FND pesticides.
11. Please include LC-MS/MS data regarding the degradation of TMX and FND pesticides, as authors claim that there were no toxic-by products formed during the process. List out the compounds identified through MS/MS. Only one compound each for TMX and FND was mentioned in Lines 205-206.
12. Discuss and Compare the results obtained using TiO2 photocatalyst with other photocatalysts, as literature is available.
13. Prove a schematic representation of possible mechanism involved in the degradation of TMX and FND pesticides based on the experimental results obtained.
Minor
1. Line 20: TiO2; 2 should be sub script (check throughout the manuscript)
2. Line 36: “usage improves the quality and augments and productivity of cereals”. Line does not make any sense, please re-write it.
3. Line 40: “eventually achieving to accumulate in soils and groundwater” replace with “eventually accumulating in soil and groundwater”.
4. Line 46: “when was firstly” replace with “when it was first”.
5. Line 54: “could harm algae or bacteria”. Does it mean it can harm any one of them or both? Rewrite the sentence accordingly.
6. Line 55: “while USA authorities are frequently reconsidering”. Does USA still using the TMX, even though it has negative impact. Provide clarification.
7. Line 57: “alternative and efficient pesticides of lower toxicity for” replace with “alternative and efficient pesticides with lower toxicity towards mammals”.
8. Line 58: rectify spelling of “floni-camid”.
9. Line 60: rectify spelling of “po-tatoes”.
10. Line 66: kindly mention the quantity or range of FND found in humans (serum and urine) and what is the acceptable or safety level for humans.
11. Figure 4a: provide clear graph (similar to Figure 4b).
12. Please go through the manuscript for grammatical and typographical mistakes.
